# Drift-Free Joint Angle Calculation Using Inertial Measurement Units without Magnetometers: An Exploration of Sensor Fusion Methods for the Elbow and Wrist

**DOI:** 10.3390/s23167053

**Published:** 2023-08-09

**Authors:** Howard Chen, Mark C. Schall, Scott M. Martin, Nathan B. Fethke

**Affiliations:** 1Industrial & Systems Engineering and Engineering Management Department, University of Alabama in Huntsville, Huntsville, AL 35899, USA; 2Department of Industrial & Systems Engineering, Auburn University, Auburn, AL 36849, USA; mark-schall@auburn.edu; 3Department of Mechanical Engineering, Auburn University, Auburn, AL 36849, USA; smm0008@auburn.edu; 4Department of Occupational & Environmental Health, The University of Iowa, Iowa City, IA 52242, USA; nathan-fethke@uiowa.edu

**Keywords:** inertial measurement units, inertial motion capture, biomechanical motion analysis, optimal estimation, biomechanics, smoothers, self-calibration, elbow, wrist

## Abstract

Joint angles of the lower extremities have been calculated using gyroscope and accelerometer measurements from inertial measurement units (IMUs) without sensor drift by leveraging kinematic constraints. However, it is unknown whether these methods are generalizable to the upper extremity due to differences in motion dynamics. Furthermore, the extent that post-processed sensor fusion algorithms can improve measurement accuracy relative to more commonly used Kalman filter-based methods remains unknown. This study calculated the elbow and wrist joint angles of 13 participants performing a simple ≥30 min material transfer task at three rates (slow, medium, fast) using IMUs and kinematic constraints. The best-performing sensor fusion algorithm produced total root mean square errors (i.e., encompassing all three motion planes) of 6.6°, 3.6°, and 2.0° for the slow, medium, and fast transfer rates for the elbow and 2.2°, 1.7°, and 1.5° for the wrist, respectively.

## 1. Introduction

Inertial motion capture systems (IMCs) are touted for their biomechanical motion analysis capabilities without the capture volume constraints imposed by traditional optical motion capture systems (OMCs). Various IMCs contain the battery and memory capacity for biomechanical motion analysis over several hours, which allows the quantification of exposures to non-neutral postures for assessing human performance in occupational ergonomics and rehabilitation applications [1,2,3]. Although commercially available IMCs are increasingly used in human motion studies [4,5], opportunities exist to (i) improve their accuracy and (ii) characterize their error characteristics. Understanding an IMC’s error characteristics and failure modes may be more advantageous to a user than potential increases in accuracy that may come from a proprietary solution [6].

IMCs rely on magnetometers to provide a directional reference around the gravitational vector. They assume that the magnetometer accurately measures the strength and direction of Earth’s local magnetic field. A major limitation of IMCs is the reduction in accuracy associated with magnetic disturbance [4,7,8,9], which occurs when the assumption of a homogenous local magnetic field is violated (e.g., indoor environments). Deviations up to 180° [8] have been reported. Without magnetometers, IMCs are limited to short-duration (i.e., minutes) full-body motion capture to avoid errors associated with gyroscopic drift [10,11] or making measurements relative to gravity (i.e., inclinometer) [3,12,13,14,15]—both approaches which limit their functionality. Various sensor fusion algorithms account for magnetic disturbances [11,16,17,18,19,20]. However, these approaches can only mitigate short-duration disturbances since they operate by discarding magnetometer measurements and relying on gyroscope measurements that drift.

IMCs can provide drift-free joint angle measurements in all three rotational movement directions without using magnetometers by incorporating linear and/or rotational motion constraints [21,22,23,24,25,26,27,28,29]. Linear motion constraints assume the linear acceleration [21,30], linear velocity [31], or linear position [22,25,26,29,32,33] at the joint center must be equal when determined from either of the two IMUs attached to adjacent body segments. A rotational constraint assumes that joints contain limited rotational motion in at least one of the three motion planes (e.g., limited abduction/adduction movement of the elbow) [27,28,34] or that joints are limited by their range of motion [24,35,36,37].

Several studies involving the lower extremity have provided promising results in leveraging motion constraints to obtain joint angles from IMUs without magnetometers. Teufl et al. [25] reported <2.5° root mean square error (RMSE) for each of the three planes of motion for the left and right knee, ankle, and hip with no observable gyroscopic drift among 28 participants performing a 6 min walk test when using linear position constraints. Similarly, Weygers et al. [21] reported <5° RMSE for each of the three motion planes of the knee with no observable gyroscopic drift among 11 participants that performed a 7 min walk test. However, it is unknown if these performance characteristics may generalize to the upper extremity, particularly at slower movement speeds, due to differences in motion dynamics. We expect that error magnitudes may be larger for the upper extremities since the IMUs are positioned more closely to the joint centers, therefore producing smaller linear acceleration magnitudes when compared to the lower extremities.

Although several studies have incorporated motion constraints when calculating joint kinematics of the upper extremity [22,23,24,38,39,40,41,42], many are still susceptible to magnetic disturbances since magnetometers are still used in their solutions. Of the studies that did not consider using magnetometers, El-Gohary et al. [23] demonstrated this capability using a continuous recording of IMU measurements that lasted approximately 2 min for four participants. Given the relatively short timeframe and use of static calibration preceding each trial, it is unknown if their approach would outperform a naïve dead-reckoned solution by integrating gyroscope measurements with respect to time. Miezal et al. [22] and Luinge et al. [28] also examined motion constraints. However, their studies considered IMU measurements from one participant, which may not generalize to other participants with different anthropometry and motion dynamics. Furthermore, the measurement timeframes were <2 min.

Sensor fusion methods within IMCs typically involve complementary filters [6,7,43,44,45,46] and Kalman filters [6,16,21,23,47,48]. A smoothing-based sensor fusion method will, theoretically, provide a more accurate solution for a non-linear system since it considers all available measurements preceding and following the solution at a given time. One commonly used smoothing-based method is the Rauch–Tung–Striebel (rts) smoother, which executes a Kalman filter forward and backward in time [49,50]. Another approach uses a maximum a posteriori estimator (map) [21,51,52,53]. This optimization-based method enables more flexibility and accuracy than an rts smoother [54]. Teufl et al. [25,26] executed their Kalman filter twice when estimating lower extremity kinematics: the first time to obtain a converged estimate of the acceleration bias, which was then used as an initial estimate in the second run. Although not strictly a smoothing-based approach, this method can only be applied once the data have been collected. Weygers et al. [21] compared a map-based smoothing approach with a multiplicative extended Kalman filter (mekf). While their smoothing-based approach performed better, the RMSE difference was <1° for each of the three movement directions. Razavi et al. [55] similarly observed an RMSE difference <1° between the smoothing of their forward–backward Kalman smoother and their Kalman filter for the pelvis and foot. To our knowledge, this comparison against a filtering-based approach has not been simultaneously made with both the map and the rts smoother, nor assessed within the context of upper extremity kinematics where sensor excitation can potentially be more limited.

In summary, the most promising studies that leveraged motion constraints to obtain joint angles from IMUs without magnetometers primarily focused on the lower extremity. Studies involving the upper extremity have contained fewer participants and shorter measurement durations. The overall objectives of this study are to characterize the (i) capability and limitations of calculating joint angles of the elbow and wrist without magnetometers at various motion speeds over >30 min among a sample of 13 participants using motion constraints and (ii) the capability of post-processed sensor fusion algorithms (rts smoother and map) to further increase measurement accuracy. In this paper, we (i) describe an rts smoother formulation of [21]. We then (ii) incorporate a rotational constraint for the elbow that limits abduction/adduction motion and a rotational constraint for the wrist to limit internal/external rotation motion that is analogous to [28]. Finally, we (iii) evaluate the effects of motion speed on the accuracy of elbow and wrist kinematics using three different sensor fusion methods (mekf, rts, map) and two different models (linear motion constraint with a degree of freedom rotational motion constraint (dof) and without a degree of freedom motion constraint (acc)), totaling six different sensor fusion algorithms (mekf-acc, mekf-dof, rts-acc, rts-dof, map-acc, map-dof). We hypothesize that the measurement accuracy for a given sensor fusion algorithm will increase with (i) increased motion speed, (ii) the use of smoothers for sensor fusion, and (iii) the addition of the rotational constraint. We further hypothesize that the map sensor fusion method will provide higher measurement accuracy compared to the rts sensor fusion method and the wrist joint will produce lower error magnitudes compared to the elbow joint.

## 2. Materials and Methods

### 2.1. Definitions, Notations, and Joint Angle Calculation Process

#### 2.1.1. Quaternion Definition

We define the quaternion vector $\vec{q}$ as Hamiltonian quaternion with a 4 × 1 column vector consisting of the scalar term $q_{0}$ for the first element and the vector term $\vec{q}$, for the subsequent three elements (1):(1)$$\vec{q}={\left[\begin{array}{cccc} q_{0} & q_{1} & q_{2} & q_{3} \end{array}\right]}^{T}=\left[\begin{array}{c} q_{0}\\ \vec{q} \end{array}\right]$$

Given quaternion vectors ${\vec{q}}_{a}$ and ${\vec{q}}_{b}$, the quaternion product ⊗ can be calculated as follows:(2)$${\vec{q}}_{a}⊗{\vec{q}}_{b}={{\vec{q}}_{a}}^{L}{\vec{q}}_{b}={{\vec{q}}_{b}}^{R}{\vec{q}}_{a}$$
(3)$${{\vec{q}}_{a}}^{L}=\left[\begin{array}{cc} {q_{a}}_{0} & -{\vec{q}}_{a}^{T}\\ {\vec{q}}_{b} & {q_{a}}_{0}I_{3}+{\left[{\vec{q}}_{a}\right]}_{\times } \end{array}\right]$$
(4)$${{\vec{q}}_{b}}^{R}=\left[\begin{array}{cc} {q_{b}}_{0} & -{\vec{q}}_{b}^{T}\\ {\vec{q}}_{b} & {q_{b}}_{0}I_{3}-{\left[{\vec{q}}_{b}\right]}_{\times } \end{array}\right]$$
where ${\left[\cdot \right]}_{\times }$ is the skew-symmetric operator and $I_{n}$ is an n × n identity matrix. The skew-symmetric operator for vector $\vec{u}$ is defined as:(5)$${\left[\vec{u}\right]}_{\times }=\left[\begin{array}{ccc} 0 & -u_{z} & u_{y}\\ u_{z} & 0 & -u_{x}\\ -u_{y} & u_{x} & 0 \end{array}\right]$$
${\vec{q}}^{GL}$ will provide a rotation from local coordinate Frame *L* to global coordinate Frame *G*, where ${\vec{q}}^{c}$ is the quaternion conjugate of $\vec{q}$, ${\vec{x}}^{L}$ is a vector in Frame *L*, and ${\vec{x}}^{G}$ is its corresponding vector in Frame *G*.
(6)$$\left[\begin{array}{c} 0\\ {\vec{x}}^{G} \end{array}\right]={\vec{q}}^{GL}⊗\left[\begin{array}{c} 0\\ {\vec{x}}^{L} \end{array}\right]⊗{\left({\vec{q}}^{GL}\right)}^{c}$$
(7)$${\vec{q}}^{c}=\left[\begin{array}{c} q_{0}\\ -\vec{q} \end{array}\right]$$

${\vec{q}}^{GL}$ can be converted into its equivalent rotation matrix $R^{GL}\;$ as follows:(8)$$R^{GL}=\left({\left(q_{0}^{GL}\right)}^{2}-{\left({\vec{q}}^{GL}\right)}^{T}{\vec{q}}^{GL}\right)I_{3}+2{\vec{q}}^{GL}{\left({\vec{q}}^{GL}\right)}^{T}+2q_{0}^{GL}{\left[{\vec{q}}^{GL}\right]}_{\times }$$
and can be subsequently used to obtain ${\vec{x}}^{G}$ from ${\vec{x}}^{L}$.
(9)$${\vec{x}}^{G}=R^{GL}{\vec{x}}^{L}\;$$

#### 2.1.2. Coordinate Frame Notations and Definitions

We define *G* as the global frame, *I* as the IMU local frame, and *S* as the local frame of a given body segment. We define ${\vec{r}}_{i}$ as the distance from the joint center to the position of the IMU in Frame *I*. $R^{GI}$ provides the orientation of IMU relative to Frame *G*, $R^{GS}$ provides the orientation of its underlying body segment relative to Frame *G*, and $R^{IS}$ is the offset between the IMU local frame and the local frame of its underlying body segment (Figure 1).

#### 2.1.3. Joint Angle Calculation Process

The process of calculating joint angles from IMU measurements is shown in Figure 2. Given two IMUs attached to adjacent body segments, i=1 denotes the proximal segment and i=2 denotes the distal segment. We determine the joint angles from angular velocity $\left({\vec{\omega }}_{t,i}\right)$ and linear acceleration $\left({\vec{a}}_{t,i}\right)$ measurements from the gyroscope and accelerometer contained within each IMU attached to a given body segment by first calculating ${\vec{r}}_{i}$ using the translational alignment process. Specifically, ${\vec{r}}_{i}$ is solved through optimization by minimizing the difference in linear acceleration vector magnitude of the proximal and distal segments at the joint center (Section 2.3.1). The orientation at time *t* of the IMU attached to the proximal body segment $\left(R_{t,1}^{GI}\right)$ and the orientation of the IMU attached to the distal body segment $\left(R_{t,2}^{GI}\right)$ are subsequently calculated from ${\vec{r}}_{1}$, ${\vec{a}}_{1}$, ${\vec{\omega }}_{1}$, and ${\vec{r}}_{2}$, ${\vec{a}}_{2}$, ${\vec{\omega }}_{2}$ using either mekf-acc, rts-acc, or map-acc (Section 2.2). Rotational alignment $R_{1}^{IS}$ and $R_{2}^{IS}$ are subsequently calculated using $R_{t,1}^{GI}$, $R_{t,2}^{GI}$ and ${\vec{\omega }}_{1}$, ${\vec{\omega }}_{2}$ by either minimizing abduction/adduction (ab/ad) for the elbow joint or internal/external rotation (in/ex) for the wrist joint (Section 2.3.2). $R_{t,1}^{GS}$, $R_{t,2}^{GS}$ can subsequently be calculated using either of the six sensor fusion algorithms (Section 2.2) given ${\vec{r}}_{1}$, ${\vec{a}}_{1}$, ${\vec{\omega }}_{1}$, $R_{1}^{IS}$ and ${\vec{r}}_{2}$, ${\vec{a}}_{2}$, ${\vec{\omega }}_{2}$, $R_{2}^{IS}$. Finally, joint angle can be calculated from $R_{t,1}^{GS}$ and $R_{t,2}^{GS}$. Note that an IMU must be attached to the upper arm and forearm to obtain elbow joint angles and to the hand and forearm to obtain wrist joint angles. Depending on the joint angle, the rotational constraint limits either elbow ab/ad or wrist in/ex. Our source code is provided as Appendix A.

### 2.2. Sensor Fusion

The combination of the sensor fusion method (e.g., mekf, rts, map) and kinematic constraint (linear acceleration constraint, linear acceleration and degree of freedom-based rotational constraint) provided six different sensor fusion algorithms: mekf-acc, rts-acc, map-acc, mekf-dof, rts-dof, map-dof. All sensor fusion algorithms presented consider the following states (8 × 1 vector):(10)$$\hat{x_{t}}=\left[\begin{array}{c} {\tilde{q}}_{t,1}^{GS}\\ {\tilde{q}}_{t,2}^{GS} \end{array}\right]$$
where ${\tilde{q}}_{t,i}^{GI}$ is the segment orientation estimated by a given sensor fusion algorithm. Since the quaternion orientation vector is overconstrained, instead of estimating ${\tilde{q}}_{t,i}^{GI}$ directly, the sensor fusion algorithms presented will instead estimate orientation deviations ${\hat{\eta }}_{t}$, which is a 6 × 1 vector containing the orientation deviation for the proximal segment (${\hat{\eta }}_{t,1}$) and distal segment (${\hat{\eta }}_{t,2}$).
(11)$$\delta \hat{x_{t}}={\hat{\eta }}_{t}=\left[\begin{array}{c} {\hat{\eta }}_{t,1}\\ {\hat{\eta }}_{t,2} \end{array}\right]$$

The orientation deviation is subsequently used to correct the orientation obtained by integrating the gyroscope. ${\hat{\eta }}_{t}$ is subsequently set to zero. The details of their implementation are shown in subsequent sections.

#### 2.2.1. Multiplicative Extended Kalman Filter

The mekf was obtained from [21], which used a linear constraint assuming that the acceleration at the joint center is identical when calculated using acceleration measurements from either IMU. Equations (12)–(26) assume solely a link constraint. Equations (27)–(34) provide the extensions to include the rotational constraint to limit elbow ab/ad and in/ex. These methods assume the IMU is already aligned with the underlying body segment.

Spatial orientation is obtained by first integrating $\vec{\omega }$ with respect to time, where Δt is the sampling period.
(12)$${\tilde{q}}_{t|t-1,i}^{GS}={\tilde{q}}_{t-1|t-1,i}^{GS}⊗\left[\begin{array}{c} 1\\ 0.5{\vec{\omega }}_{t-1,i}\Delta t \end{array}\right]$$

Note that the Kalman filter state transition equation is unnecessary since ${\hat{\eta }}_{t}$ is zero at the start of each time update, and the inputs are assumed to be zero mean Gaussian.

The predicted estimate covariance $P_{t|t-1}$ is calculated using Equation (13).
(13)$$P_{t|t-1}=F_{t-1}P_{t-1|t-1}F_{t-1}^{T}+GQG^{T}$$

Gyroscope orientation deviation transition matrix F, gyroscope noise Jacobian *G*, and gyroscope noise covariance *Q* are calculated as follows:(14)$$F_{t-1}=\left[\begin{array}{cc} I_{3}-{\left[{\vec{\omega }}_{t-1,1}\Delta t\right]}_{\times } & 0_{3}\\ 0_{3} & I_{3}-{\left[{\vec{\omega }}_{t-1,2}\Delta t\right]}_{\times } \end{array}\right]$$
(15)$$G=I_{6}\Delta t$$
(16)$$Q=I_{6}\sigma_{\omega }^{2}$$
where $\sigma_{\omega }^{2}$ is the gyroscope white noise and $0_{n}$ is an n × n matrix of zeros.

Kalman gain $K_{t}$ is calculated as follows:(17)$$K_{t}=P_{t|t-1}H_{t}^{T}{\left(H_{t}P_{t|t-1}H_{t}^{T}+R\right)}^{-1}$$
with measurement Jacobian H defined as:(18)$$H_{t,acc}=\left({\tilde{R}}_{t,1}^{GS}{\left[{\tilde{a}}_{t,1}\right]}_{\times }-{\tilde{R}}_{t,2}^{GS}{\left[{\tilde{a}}_{t,2}\right]}_{\times }\right)$$
(19)$${\tilde{a}}_{t,i}={\vec{a}}_{t,i}-D_{t,i}{\vec{r}}_{i}$$
(20)$$D_{t,i}=\left[\begin{array}{ccc} -\omega_{y,t,i}^{2}-\omega_{z,t,i}^{2} & \omega_{x,t,i}\omega_{y,t,i}-{\dot{\omega }}_{z,t,i} & {\dot{\omega }}_{y,t,i}+\omega_{x,t,i}\omega_{z,t,i}\\ {\dot{\omega }}_{z,t,i}+\omega_{x,t,i}\omega_{y,t,i} & -\omega_{x,t,i}^{2}-\omega_{z,t,i}^{2} & \omega_{y,t,i}\omega_{z,t,i}-{\dot{\omega }}_{x,t,i}\\ \omega_{x,t,i}\omega_{z,t,i}-{\dot{\omega }}_{y,t,i} & {\dot{\omega }}_{x,t,i}+\omega_{y,t,i}\omega_{z,t,i} & -\omega_{x,t,i}^{2}-\omega_{y,t,i}^{2} \end{array}\right]$$
where ${\tilde{R}}_{t,i}^{GS}$ is the corresponding rotation matrix of ${\tilde{q}}_{t,i}^{GS}$, $D_{t,i}{\vec{r}}_{i}$ is the lever arm equation, and $\dot{\omega }$ is its derivative calculated using a 3rd order approximation [56]:(21)$${\dot{\omega }}_{t}=\frac{\omega_{t-2}-8\omega_{t-1}+8\omega_{t+1}-\omega_{t+2}}{12\Delta t}$$

$H_{t}=H_{t,acc}$ with solely a link constraint.

Measurement noise covariance matrix R is defined as follows:(22)$$R=I_{3}\sigma_{acc}^{2}$$
where $\sigma_{acc}^{2}$ is the acceleration constraint noise.

${\hat{\eta }}_{t}$ is calculated as follows:(23)$${\hat{\eta }}_{t}=K_{t}{\delta z}_{t}$$
(24)$${\delta z}_{t,acc}={\tilde{R}}_{t,1}^{GS}{\tilde{a}}_{t,1}-{\tilde{R}}_{t,2}^{GS}{\tilde{a}}_{t,2}$$

${\delta z}_{t}={\delta z}_{t,acc}\;$ with solely a link constraint and the updated estimate covariance $P_{t|t}$ is calculated as
(25)$$P_{t|t}=\left(I_{6}-K_{t}H_{t}\right)P_{t|t-1}$$

Finally, $\eta_{t,i}$ is used to correct for ${\tilde{q}}_{t|t-1,i}^{GI}$ as follows:(26)$${\tilde{q}}_{t|t,i}^{GI}={\tilde{q}}_{t|t-1,i}^{GI}⊗\left[\begin{array}{c} 1\\ 0.5{\hat{\eta }}_{t,i} \end{array}\right]$$

##### Addition of Rotational Constraint

The degree of freedom rotational constraint for the elbow (27) adopted from [33] was added as an additional measurement update to limit elbow ab/ad.
(27)$${\delta z}_{t,el\_dof}=-{\vec{u}}_{3}^{T}{\left({\tilde{R}}_{t,1}^{GS}\right)}^{T}{\tilde{R}}_{t,2}^{GS}{\vec{u}}_{1}$$

The corresponding Jacobian is calculated using (28).
(28)$$H_{t,el\_dof}={\left[\begin{array}{c} -{\vec{u}}_{2}^{T}{\left({\tilde{R}}_{t,1}^{GS}\right)}^{T}{\tilde{R}}_{t,2}^{GS}{\vec{u}}_{1}\\ {\vec{u}}_{1}^{T}{\left({\tilde{R}}_{t,1}^{GS}\right)}^{T}{\tilde{R}}_{t,2}^{GS}{\vec{u}}_{1}\\ 0\\ 0\\ -{\vec{u}}_{3}^{T}{\left({\tilde{R}}_{t,2}^{GS}\right)}^{T}{\tilde{R}}_{t,1}^{GS}{\vec{u}}_{3}\\ {\vec{u}}_{2}{\left({\tilde{R}}_{t,2}^{GS}\right)}^{T}{\tilde{R}}_{t,1}^{GS}{\vec{u}}_{3} \end{array}\right]}^{T}$$

Unit vectors $u_{1}$, $u_{2}$, and $u_{3}$ are defined as:(29)$$\begin{array}{c} {\vec{u}}_{1}={\left[\begin{array}{ccc} 1 & 0 & 0 \end{array}\right]}^{T}\\ {\vec{u}}_{2}={\left[\begin{array}{ccc} 0 & 1 & 0 \end{array}\right]}^{T}\\ {\vec{u}}_{3}={\left[\begin{array}{ccc} 0 & 0 & 1 \end{array}\right]}^{T} \end{array}$$

Similarly, the degree of freedom rotational constraint for the wrist (30) was added as an additional measurement update to limit wrist in/ex.
(30)$${\delta z}_{t,wr\_dof}=-{\vec{u}}_{2}^{T}{\left({\tilde{R}}_{t,1}^{GS}\right)}^{T}{\tilde{R}}_{t,2}^{GS}{\vec{u}}_{3}$$

The corresponding Jacobian is calculated using (31).
(31)$$H_{t,wr\_dof}={\left[\begin{array}{c} {\vec{u}}_{3}^{T}{\left({\tilde{R}}_{t,1}^{GS}\right)}^{T}{\tilde{R}}_{t,2}^{GS}{\vec{u}}_{3}\\ 0\\ -{\vec{u}}_{1}^{T}{\left({\tilde{R}}_{t,1}^{GS}\right)}^{T}{\tilde{R}}_{t,2}^{GS}{\vec{u}}_{3}\\ -{\vec{u}}_{2}^{T}{\left({\tilde{R}}_{t,2}^{GS}\right)}^{T}{\tilde{R}}_{t,1}^{GS}{\vec{u}}_{2}\\ {\vec{u}}_{1}{\left({\tilde{R}}_{t,2}^{GS}\right)}^{T}{\tilde{R}}_{t,1}^{GS}{\vec{u}}_{2}\\ 0 \end{array}\right]}^{T}$$

With the addition of the rotation constraint, $\delta z_{t}$, $H_{t}$, and R become
(32)$${\delta z}_{t}=\left[\begin{array}{c} \delta z_{t,acc}\\ \delta z_{t,dof} \end{array}\right]$$
(33)$$H_{t}=\left[\begin{array}{c} H_{t,acc}\\ H_{t,dof} \end{array}\right]$$
(34)$$R=\left[\begin{array}{cc} I_{3}\sigma_{acc}^{2} & 0_{3}\\ 0 & \sigma_{dof}^{2} \end{array}\right]$$
where dof corresponds to either el_dof or wl_dof, depending on the joint of interest.

#### 2.2.2. Rauch–Tung–Striebel Smoother

The rts first requires running the mekf with values of $P_{t|t-1}$, $F_{t-1}$, $P_{t|t}$, ${\hat{\eta }}_{t}$, and ${\tilde{q}}_{t|t,i}^{GS}$ from (13), (14), (23), (25), and (26), respectively, to be stored. These values are used when computing the smoothed orientation deviation from the rts smoother ${\hat{\eta }}_{t|n}$, its associated covariance $P_{t|n}$, and the smoothed orientation ${\tilde{q}}_{t|n,i}^{GI}$ by executing (35) through (38) from t = n − 1 to t = 1.
(35)$$C_{t}=P_{t|t}F_{t}^{T}P_{t+1|t}^{-1}$$
(36)$${\hat{\eta }}_{t|n}={\hat{\eta }}_{t}+C_{t}{\hat{\eta }}_{t+1|n}$$
(37)$$P_{t|n}=P_{t|t}+C_{t}\left(P_{t+1|n}-P_{t+1|t}\right)C_{t}^{T}$$
(38)$${\tilde{q}}_{t|n,i}^{GI}={\tilde{q}}_{t|t,i}^{GI}⊗\left[\begin{array}{c} 1\\ 0.5{\hat{\eta }}_{t|n,i} \end{array}\right]$$

Since the iterations are executed ‘backwards’, $P_{t+1|t}$ is used instead of $P_{t|t-1}$. ${\hat{\eta }}_{t+1|n}$ is initialized to the value of ${\hat{\eta }}_{t}$ at t = n and $P_{t+1|n}$ is initialized to the value of $P_{t|t}$ at t = n, where n is the number of measurements. Consistent with the mekf, (11) is used to calculate ${\hat{\eta }}_{t|n,i}$ from ${\hat{\eta }}_{t|n}$. More information on implementation of rts can be obtained from [50].

#### 2.2.3. Maximum a Posteriori

The map sensor fusion algorithm with a linear acceleration constraint from [21] was implemented. Levenberg–Marquardt was used as the solver instead of Gauss–Newton for convergence stability. Generalized implementation information on map can be obtained from [54]. In brief, map for this application considers the following objective function:(39)$${\hat{\eta }}_{1:2n}=\arg\min\left({\hat{\eta }}_{1:2n}\right)\sum_{i=1}^{2}\left({\left\|e_{init,i}\right\|}_{\Sigma_{init}^{-1}}^{2}+\sum_{t=2}^{N}{\left\|e_{t,\omega ,i}\right\|}_{\Sigma_{\omega }^{-1}}^{2}\right)+\sum_{t=1}^{N}{\left\|e_{t,acc}\right\|}_{\Sigma_{acc}^{-1}}^{2}$$
where $e_{init,i}$, $e_{t,\omega ,i}$, $e_{t,acc}$ are the residuals associated with the initial orientation, orientation derived from gyroscope measurements, and the linear acceleration constraint, respectively, and $\Sigma_{init}$, $\Sigma_{\omega }$, $\Sigma_{acc}$ are the corresponding noise covariance matrices. Note that the minimization in (39) assumes that the residuals are Gaussian with zero mean. Minimizing the residual is equivalent to maximizing the stochastic likelihood. This objective function was solved by iterating the Levenberg–Marquardt solver until the solution converged. The orientation deviation at iteration k is calculated by iterating the following equations:(40)$$\eta_{1:2n}^{\left(k\right)}=-{\left(J^{T}WJ+\lambda diag\left(J^{T}WJ\right)\right)}^{-1}J^{T}We$$
(41)$${\tilde{q}}_{1:n,i}^{GS,(k)}={\tilde{q}}_{1:n,i}^{GS,(k-1)}⊗\left[\begin{array}{c} 1\\ 0.5\eta_{1:n,i}^{\left(k\right)} \end{array}\right]$$
where λ is the damping parameter, $\eta_{1:n,1}^{\left(k\right)}$, ${\tilde{q}}_{1:n,1}^{GS,(k)}$ correspond to $\eta_{1:n}^{\left(k\right)}$, ${\tilde{q}}_{1:n}^{GS,(k)}$, respectively, $\eta_{1:n,2}^{\left(k\right)}$, ${\tilde{q}}_{1:n,2}^{GS,(k)}$ correspond to $\eta_{n+1,2n}^{\left(k\right)}$, ${\tilde{q}}_{n+1,2n}^{GS,(k)}$, and W is a diagonal matrix containing the weights, as follows:(42)$$W=diag(\left\{1/\sigma_{init,1}^{2},3\right\},\left\{1/\sigma_{\omega ,1}^{2},3\left(n-1\right)\right\},\left\{1/\sigma_{init,2}^{2},3\right\},\left\{1/\sigma_{\omega ,2}^{2},3\left(n-1\right)\right\},\left\{1/\sigma_{link}^{2},3n\right\}$$
where diag(·) is the diagonal operator, and {a,  m}  will repeat value a m times.

The error function (e) is defined as follows:(43)$$e={\left[\begin{array}{ccccc} e_{init,1}^{T} & e_{\omega ,1}^{T} & e_{init,2}^{T} & e_{\omega ,2}^{T} & e_{acc}^{T} \end{array}\right]}^{T}$$
(44)$$e_{init,i}=2{\;\log}_{q}\left(q_{init}^{c}⊗{\hat{q}}_{1,i}\right)$$
(45)$$e_{\omega ,i}=2{\;\log}_{q}\left(q_{1:n-1,i}^{c}⊗{\hat{q}}_{2:n,i}\right)-{\vec{\omega }}_{1:n-1,i}$$
where ${log}_{q}\left(\vec{q}\right)\approx \vec{q}$, $e_{acc}=\delta z_{acc}$, which is calculated with (24), and $q_{init}$ is the initial orientation value and ${\hat{q}}_{1,i}$ is the estimated value of ${\hat{q}}_{i}$ at *t* = 1. Note $e_{init,i}$ has a dimension of 3 × 1, $e_{\omega ,i}$ has a dimension of 3(*n* − 1) × 1, and $e_{link}$ has a dimension of 3n × 1.

The corresponding Jacobian matrix J is defined as
(46)$$J={\left[\begin{array}{ccccc} {\frac{de_{init,1}}{d\eta_{1:n,1}\;}}^{T} & {\frac{de_{\omega ,1}}{d\eta_{2:n,1}}}^{T} & {\frac{de_{init,2}}{d\eta_{1:n,2}}}^{T} & {\frac{de_{\omega ,2}}{dn_{2:n,2}}}^{T} & {\frac{de_{acc}}{d\eta_{1:n}}}^{T} \end{array}\right]}^{T}$$
where $\frac{de_{init,i}}{d\eta_{1:n}}$, $\frac{de_{\omega ,i}}{d\eta_{1:n}}$, and $\frac{de_{acc}}{d\eta_{1:n}}$ are the partial derivatives of $e_{init,i}$, $e_{\omega ,i}$, and $e_{acc}$ with respect to $\eta_{1:2n}$.

These 3 × 3 matrices at time t are calculated as follows:(47)$$\frac{de_{init,i}}{d\eta_{1,i}\;}=\frac{dlog_{q}\left(q\right)}{dq}{\left(q_{init,i}⊗{\tilde{q}}_{1,i}^{GS}\right)}^{L}\frac{d{exp}_{q}\left(\eta_{1,i}^{b}\right)}{\eta_{1,i}^{b}}$$
(48)$$\frac{de_{\omega ,i}}{d\eta_{t,i}\;}=\frac{1}{\Delta t}\frac{dlog_{q}\left(q\right)}{dq}{\left({{\tilde{q}}_{t,i}^{GS}}^{c}⊗{\tilde{q}}_{t+1,i}^{GS}\right)}^{R}\frac{d{exp}_{q}{\left(\eta_{t,i}^{b}\right)}^{c}}{\eta_{t,i}^{b}}\frac{d{exp}_{q}\left(\eta_{t,i}^{b}\right)}{\eta_{t,i}^{b}}$$
(49)$$\frac{de_{\omega ,i}}{d\eta_{t+1,i}\;}=\frac{1}{\Delta t}\frac{dlog_{q}\left(q\right)}{dq}{\left({{\tilde{q}}_{t,i}^{GS}}^{c}⊗{\tilde{q}}_{t+1,i}^{GS}\right)}^{L}\frac{d{exp}_{q}\left(\eta_{t+1,i}^{b}\right)}{\eta_{t+1,i}^{b}}$$
(50)$$\frac{de_{\omega ,i}}{d\eta_{t+1,i}\;}=\frac{1}{\Delta t}\frac{dlog_{q}\left(q\right)}{dq}{\left({{\tilde{q}}_{t,i}^{GS}}^{c}⊗{\tilde{q}}_{t+1,i}^{GS}\right)}^{L}\frac{d{exp}_{q}\left(\eta_{t+1,i}^{b}\right)}{\eta_{t+1,i}^{b}}$$
(51)$$\frac{de_{link,1}}{d\eta_{t,1}\;}={-\tilde{R}}_{t,1}^{GS}{\left[{\tilde{a}}_{t,1}\right]}_{\times }$$
(52)$$\frac{de_{link,2}}{d\eta_{t,2}\;}={\tilde{R}}_{t,2}^{GS}{\left[{\tilde{a}}_{t,2}\right]}_{\times }$$
where
(53)$$\frac{dlog_{q}\left(q\right)}{dq}=\left[\begin{array}{cccc} 0 & 1 & 0 & 0\\ 0 & 0 & 1 & 0\\ 0 & 0 & 0 & 1 \end{array}\right]$$
(54)$$\frac{d{\left({exp}_{q}\left(\eta \right)\right)}^{c}}{dexp\;\eta }=\left[\begin{array}{cccc} 1 & 0 & 0 & 0\\ 0 & -1 & 0 & 0\\ 0 & 0 & -1 & 0\\ 0 & 0 & 0 & -1 \end{array}\right]$$
(55)$$\frac{d{exp}_{q}\eta }{d\eta }=\left[\begin{array}{ccc} 0 & 0 & 0\\ 1 & 0 & 0\\ 0 & 1 & 0\\ 0 & 0 & 1 \end{array}\right]$$

##### Addition of Rotational Constraint

The objective function, error function, its corresponding Jacobian, and the weight matrix with the addition of the rotation constraint are as follows:(56)$$\begin{array}{c} {\hat{\eta }}_{1:n}=\arg\min\left({\hat{\eta }}_{1:n}\right)\sum_{i=1}^{2}\left({\left\|e_{init,i}\right\|}_{\Sigma_{init}^{-1}}^{2}+\sum_{t=2}^{N}{\left\|e_{\omega ,t,i}\right\|}_{\Sigma_{\omega }^{-1}}^{2}\right)\\ \;\;\;\;\;+\sum_{t=1}^{N}{\left\|e_{acc,t}\right\|}_{\Sigma_{acc}^{-1}}^{2}+\sum_{t=1}^{N}{\left\|e_{dof,t}\right\|}_{\Sigma_{dof}^{-1}}^{2} \end{array}$$
(57)$$e={\left[\begin{array}{cccccc} e_{init,1}^{T} & e_{\omega ,1}^{T} & {e_{init,2}^{T}}^{T} & e_{\omega ,1}^{T} & e_{acc}^{T} & e_{dof}^{T} \end{array}\right]}^{T}$$
(58)$$J={\left[\begin{array}{cccccc} {\frac{de_{init,1}}{d\eta_{1:n,1}\;}}^{T} & {\frac{de_{\omega ,1}}{d\eta_{2:n,1}}}^{T} & {\frac{de_{init,2}}{d\eta_{1:n,2}}}^{T} & {\frac{de_{\omega ,2}}{dn_{2:n,2}}}^{T} & {\frac{de_{acc}}{d\eta_{1:n}}}^{T} & {\frac{de_{dof}}{d\eta_{1:n}}}^{T} \end{array}\right]}^{T}$$
(59)$$W=diag(\left\{1/\sigma_{init,1}^{2},3\right\},\left\{1/\sigma_{\omega ,1}^{2},3\left(n-1\right)\right\},\left\{1/\sigma_{init,2}^{2},3\right\},\left\{1/\sigma_{\omega ,2}^{2},3\left(n-1\right)\right\},\left\{1/\sigma_{acc}^{2},3n\right\},\left\{1/\sigma_{dof}^{2},n\right\})$$

For the elbow,
(60)$$e_{t,dof}={\vec{u}}_{3}^{T}{\left({\tilde{R}}_{t,1}^{GS}\right)}^{T}{\tilde{R}}_{t,2}^{GS}{\vec{u}}_{1}$$
(61)$$\begin{array}{c} \frac{de_{t,dof}}{d\eta_{t,1}}={\left[\begin{array}{c} -{\vec{u}}_{2}^{T}{\left(R_{t,1}^{GS}\right)}^{T}\;R_{t,2\;}^{Gs}{\vec{u}}_{1}\\ {\vec{u}}_{1}^{T}{\left(R_{t,1}^{GS}\right)}^{T}R_{t,2\;}^{GS}{\vec{u}}_{1}\\ 0 \end{array}\right]}^{T}\\ \frac{de_{t,dof}}{d\eta_{t,2}}={\left[\begin{array}{c} 0\\ {\vec{u}}_{3}^{T}{\left(R_{t,2}^{GS}\right)}^{T}\;R_{t,1}^{GS}{\vec{u}}_{3}\\ {-\vec{u}}_{2}{\left(R_{t,2}^{GS}\right)}^{T}\;R_{t,1}^{GS}{\vec{u}}_{3} \end{array}\right]}^{T} \end{array}$$

For the wrist,
(62)$$e_{t,dof}={\vec{u}}_{3}^{T}{\left({\tilde{R}}_{t,1}^{GS}\right)}^{T}{\tilde{R}}_{t,2}^{GS}{\vec{u}}_{1}$$
(63)$$\begin{array}{c} \frac{de_{t,dof}}{d\eta_{t,1}}={\left[\begin{array}{c} {\vec{u}}_{3}^{T}{\left({\tilde{R}}_{t,1}^{GS}\right)}^{T}{\tilde{R}}_{t,2}^{GS}{\vec{u}}_{3}\\ 0\\ -{\vec{u}}_{1}^{T}{\left({\tilde{R}}_{t,1}^{GS}\right)}^{T}{\tilde{R}}_{t,2}^{GS}{\vec{u}}_{3} \end{array}\right]}^{T}\\ \frac{de_{t,dof}}{d\eta_{t,2}}={\left[\begin{array}{c} -{\vec{u}}_{2}^{T}{\left({\tilde{R}}_{t,2}^{GS}\right)}^{T}{\tilde{R}}_{t,1}^{GS}{\vec{u}}_{2}\\ {\vec{u}}_{1}{\left({\tilde{R}}_{t,2}^{GS}\right)}^{T}{\tilde{R}}_{t,1}^{GS}{\vec{u}}_{2}\\ 0 \end{array}\right]}^{T} \end{array}$$

### 2.3. IMU-to-Segment Alignment

#### 2.3.1. Translational Alignment

The distance from the joint center to each IMU (${\vec{r}}_{i}$) is calculated using the method from [56], which solves these distances by minimizing the acceleration magnitude at the joint center. The cost function is as follows:(64)$$e{\left(\Phi \right)}_{t}=\left\|{\vec{a}}_{t,1}-D_{t,1}{\vec{r}}_{1}\right\|-\left\|{\vec{a}}_{t,2}-D_{t,2}{\vec{r}}_{2}\right\|$$
where $K_{t,i}$ is defined in (20).

The parameter vector Φ is defined as
(65)$$\Phi =\left[\begin{array}{cccccc} r_{x,1} & r_{y,1} & r_{z,1} & r_{x,2} & r_{y,2} & r_{z,2} \end{array}\right]$$
and
(66)$$J_{t}=\left[\begin{array}{cc} \frac{d\left\|{\vec{a}}_{t,1}-D_{t,1}r_{1}\right\|}{dr_{1}} & \frac{d\left\|{\vec{a}}_{t,2}-D_{t,2}r_{2}\right\|}{dr_{2}} \end{array}\right]$$
(67)$$\frac{d\left\|{\vec{a}}_{t,1}-D_{t,1}r_{1}\right\|}{dr_{1}}=-\frac{{\left({\vec{a}}_{t,1}-K_{t,1}r_{1}\right)}^{T}D_{t,1}}{\left\|{\hat{a}}_{t,1}-K_{t,1}r_{1}\right\|}$$
(68)$$\frac{d\left\|{\vec{a}}_{t,2}-D_{t,2}r_{2}\right\|}{Dr_{2}}=\frac{{\left({\vec{a}}_{t,2}-D_{t,2}r_{2}\right)}^{T}D_{t,2}}{\left\|{\vec{a}}_{t,2}-D_{t,2}r_{2}\right\|}$$

Φ is solved by iterating the following until convergence.
(69)$$\Phi^{\left(k\right)}=\Phi^{\left(k-1\right)}-pinv(J)e{\left(\Phi \right)}_{k}$$
where pinv(·) is the Moore–Penrose pseudoinverse and $\left\|\cdot \right\|$ is the 2-norm operator.

#### 2.3.2. Rotational Alignment

IMUs were rotationally aligned to the underlying upper arm and forearm body segments using the extrinsic calibration method from [27]. Modifications were made to omit the use of magnetometer measurements. Specifically, orientation measurements were obtained from a sensor fusion algorithm that used the linear acceleration constraint to omit the use of magnetometer measurements. The heading error parameters in [27] used to mitigate magnetometer disturbance were therefore excluded. Additionally, parameters were solved using Levenberg–Marquardt instead of Gauss–Newton to improve reliability.

The joint axes $j_{1}$ and $j_{2}$ were parameterized as
(70)$$j_{i}={\left[\begin{array}{ccc} sin{(\theta }_{i})cos{(\psi }_{i}) & \sin\left(\theta_{i}\right)\sin\left(\psi_{i}\right) & \cos\left(\theta_{i}\right) \end{array}\right]}^{T}$$

An alternative parameterization is as follows:(71)$$j_{i}={\left[\begin{array}{ccc} cos{(\theta }_{i}) & \sin\left(\theta_{i}\right)\sin\left(\psi_{i}\right) & \cos\left(\psi_{i}\right) \end{array}\right]}^{T}$$
where $\theta_{i}$ and $\psi_{i}$ are scalar values that define angles in the spherical coordinates, which are used to ensure $j_{i}$ is a unit vector. To alleviate singularities, the two parameterizations for $j_{i}$ were switched from one to another whenever $\left|\sin\left(\theta_{i}\right)\right|<0.5$.

The 4 × 1 parameter vector Φ and its corresponding error function $e{\left(\Phi \right)}_{k}$ are defined as follows:(72)$$\Phi ={\left[\begin{array}{cccc} \theta_{1} & \phi_{1} & \theta_{2} & \phi_{2} \end{array}\right]}^{T}$$
(73)$$e{\left(\Phi \right)}_{t}={(R}_{t,1\;}^{GI}\omega_{1}-R_{t,2\;}^{GI}\omega_{2})\cdot (R_{t,1\;}^{GI}j_{1}\times R_{t,2\;}^{GI}j_{2})$$
where · is the vector dot product, × is the vector cross product, and $R_{t,i}^{GI}$ is the orientation of the IMU (local Frame I) relative to the global Frame G. θ and ϕ are parameters used to define the direction of a unit vector in Cartesian space.

Φ is solved by iterating the following until convergence.
(74)$$\Phi^{\left(k\right)}=\Phi^{\left(k-1\right)}-{\left(J^{T}J+\lambda diag\left(J^{T}J\right)\right)}^{-1}J^{T}e{\left(\Phi \right)}_{k}$$
where J is the partial derivative $e{\left(\Phi \right)}_{t}$ with respect to Φ. The rotational offset from inertial Frame I to segment intermediate Frame S’ is calculated as follows:(75)$$q_{i}^{IS^{\prime }}=(\cos^{-1}\left({\left[\begin{array}{ccc} 0 & 0 & 1 \end{array}\right]}^{T}\cdot j_{i}\right),{\left[\begin{array}{ccc} 0 & 0 & 1 \end{array}\right]}^{T}\times j_{i})$$
and
(76)$$\left(\alpha ,u\right)=\left[\begin{array}{c} \cos\left(\frac{\alpha }{2}\right)\\ \sin\left(\frac{\alpha }{2}\right)u/\left\|u\right\| \end{array}\right]$$

The rotational offset from inertial frame to segment frame $\left(q_{i}^{IS}\right)$ is calculated as follows:(77)$$q_{i}^{IS}=q_{i}^{IS^{\prime }}⊗\left(\alpha_{0,i}-\alpha_{r,i},{\left[\begin{array}{ccc} 0 & 0 & 1 \end{array}\right]}^{T}\right)$$
where $\alpha_{0}$ is the measured orientation and $\alpha_{r}$ is the reference orientation for the joint segment, which we define as the orientation of a given segment at neutral posture.

Once $q_{i}^{IS}$ is obtained, gyroscope measurements $\left({\vec{\omega }}_{t,i}^{s}\right)$, accelerometer measurements $\left({\hat{a}}_{t,i}^{s}\right)$, and distances from the joint center to the adjacent IMUs ${\vec{r}}_{i}^{s}$ in the segment frame are calculated as follows.
(78)$${\vec{\omega }}_{t,i}^{s}={\left(R_{i}^{IS}\right)}^{T}{\vec{\omega }}_{t,i}$$
(79)$${\hat{a}}_{t,i}^{s}={\left(R_{i}^{IS}\right)}^{T}{\hat{a}}_{t,i}$$
(80)$${\vec{r}}_{i}^{s}={\left(R_{i}^{IS}\right)}^{T}{\vec{r}}_{i}$$
and used in the sensor fusion equations instead of ${\vec{\omega }}_{t,i}$, ${\hat{a}}_{t,i}$, and ${\vec{r}}_{i}$ to solve for spatial orientations in the segment frame. Note that $R_{i}^{IS}$ is the rotation matrix equivalent of $q_{i}^{IS}$.

Elbow orientation $q_{t,el}$ is calculated as follows:(81)$$q_{t,el}={\left(q_{t,1}^{GS}\right)}^{c}⊗q_{t,2}^{GS}$$
where *i* = 1 denotes the upper arm orientation and *i* = 2 denotes the forearm orientation.

The rotational alignment for the wrist was obtained by solving for the parameter vector Φ containing the orientation deviations corresponding to the IMU’s rotational alignment to the forearm and hand. The first three elements correspond to the alignment of the forearm IMU, while the last three elements correspond to the hand alignment. Specifically, the following was iterated until convergence:(82)$$\Phi^{\left(k+1\right)}=-\lambda_{wr}{\left(J_{wr}^{T}J_{wr}\right)}^{-1}J_{wr}^{T}e_{wr}$$
(83)$$q_{1}^{IS,(k+1)}=q_{1}^{IS,(k)}⊗{\left[\begin{array}{cc} 1 & 0.5{\Phi_{1:3}^{\left(k+1\right)}}^{T} \end{array}\right]}^{T}$$
(84)$$q_{2}^{IS,(k+1)}=q_{2}^{IS,(k)}⊗{\left[\begin{array}{cc} 1 & 0.5{\Phi_{4:6}^{\left(k+1\right)}}^{T} \end{array}\right]}^{T}$$
where
(85)$$e_{wr}={\vec{u}}_{2}^{T}{\left({\tilde{R}}_{t,1}^{GI}R_{1}^{IS}\right)}^{T}({\tilde{R}}_{t,2}^{GI}R_{2}^{IS}){\vec{u}}_{3}$$
(86)$$H_{t,wr\_dof}={\left[\begin{array}{c} {\vec{u}}_{3}^{T}{\left({\tilde{R}}_{t,1}^{GI}R_{1}^{IS}\right)}^{T}({\tilde{R}}_{t,2}^{GI}R_{2}^{IS}){\vec{u}}_{3}\\ 0\\ -{\vec{u}}_{1}^{T}{\left({\tilde{R}}_{t,1}^{GI}R_{1}^{IS}\right)}^{T}({\tilde{R}}_{t,2}^{GI}R_{2}^{IS}){\vec{u}}_{3}\\ -{\vec{u}}_{2}^{T}{\left({\tilde{R}}_{t,1}^{GI}R_{1}^{IS}\right)}^{T}({\tilde{R}}_{t,2}^{GI}R_{2}^{IS}){\vec{u}}_{2}\\ {\vec{u}}_{1}{\left({\tilde{R}}_{t,1}^{GI}R_{1}^{IS}\right)}^{T}({\tilde{R}}_{t,2}^{GI}R_{2}^{IS}){\vec{u}}_{2}\\ 0 \end{array}\right]}^{T}$$

Note that Φ for the rotational alignment of the wrist is a 6 × 1 vector; *i* = 1 and *i* = 2 correspond to the forearm and hand segments, respectively; and $R_{i}^{IS}$ is the rotation matrix equivalent of $q_{i}^{IS}$. Equations (78)–(80) are subsequently used for angular velocity, linear acceleration, and lever arm from the IMU frame to the segment frame, respectively.

### 2.4. Data Collection

We refer to our previous work [6,7,15,57] for a comprehensive description of our data collection methods. In brief, 13 right-handed participants (11 male; mean age = 27.2 ± 6.6 years) each completed six trials of a material transfer task that involved transferring wooden dowels from a waist-height container directly in front of the participant to a shoulder-height container for one minute. The exclusion criteria included any self-reported cases of (i) physician-diagnosed MSD in the previous six months, (ii) pain during the two weeks prior to study enrollment, and (iii) history of orthopedic surgery in the upper extremity (shoulder, elbow, wrist, hand). Written informed consent was obtained prior to data collection. The study protocol was approved by the University of Iowa Institutional Review Board.

The six trials consisted of two at three different material transfer rates (15, 30, and 45 transfers per minute) dictated by a metronome. The transfer rate for each trial was randomly assigned. Each trial was followed by a five-minute rest period and a ‘practice’ period for the participant to acquaint themselves with the given transfer rate. IMUs (Opal, APDM) were attached to the dominant upper arm, forearm, and hand using Velcro^®^ straps for each participant. The IMUs were oriented with the positive x-axis parallel to the body segment and pointed distally, the positive y-axis parallel to the sagittal plane pointed forward, and the positive z-axis perpendicular to the sagittal plane pointed laterally when the participant was in the ‘I-pose’. IMU and OMC measurements were recorded at 128 and 120 Hz, respectively. An OMC (Optitrack Flex13, Naturalpoint, Corvallis, OR, USA) was used to track four reflective markers attached to each IMU’s top surface. OMC spatial orientation was calculated for each marker cluster using the manufacturer-provided software and software package (Motive: Body version 1.10.0, NaturalPoint, Inc. Corvallis, OR, USA). IMU measurements were recorded continuously and exceeded 30 min in duration for each participant.

### 2.5. Data Analysis

#### Implementation of Sensor Fusion Algorithms and IMU-to-Segment Alignment

All computation was accomplished using MATLAB (R2021a, Mathworks) except for map-acc and map-dof. The map algorithms were implemented in C++ using the Eigen 3.4.0 linear algebra library due to computational overhead. The settings for the sensor fusion algorithm are shown in Table 1 for the elbow and Table 2 for the wrist. Initial quaternion was set to [1, 0, 0, 0] for all sensor fusion algorithms. Map was iterated until either (i) local minimum, (ii) maximum iteration of 25, or (iii) tolerance of 0.01 had been reached. A dampening parameter (λ) of 1 × 10^−8^ was used.

Translation and rotational alignment were determined using the same trial with a ‘fast’ transfer rate. For both the elbow and wrist joints, the translational alignment algorithm was initialized to zeros and was executed for 20 iterations. The rotational alignment was calculated with the spatial orientation measurements provided using rts-acc with the assumption that the IMU was translationally aligned to the underlying body segment. The rotational alignment for the elbow was iterated until one of the following was reached: (i) a local minimum, (ii) a maximum iteration of 100, or (iii) a tolerance of 0.001. The λ dampening parameter was set to 1. The rotational alignment for the wrist was iterated until one of the following was reached: (i) a local minimum, (ii) a maximum iteration of 50, or (iii) a tolerance of 0.001. The λ dampening parameter was set to 0.0001.

The offset between the OMC-derived relative orientation and IMU-derived relative orientation was determined using the method described in [58] and applied to OMC measurements. This procedure was consistent with [21]. Error at a given time is defined according to [59]:
(87)$${\vec{q}}_{t,ERR}={\vec{q}}_{t,REF}⊗{\vec{q}}_{t,\mathrm{IMU}}^{c}$$
where ${\vec{q}}_{t,REF}$ is the relative orientation from the OMC with the alignment offsets added, ${\vec{q}}_{t,IMU}$ is the elbow joint angle. The OMC was aligned to the IMU using the same trial for IMU-to-segment alignment (I2S). ${\vec{q}}_{t,ERR}$ was subsequently decomposed to an Euler rotation sequence of z,y′,x″ to obtain error in fl/ex, ab/ad, and internal/external rotation (in/ex), respectively, at time t for the elbow and x-y′-z″ to obtain in/ex, fl/ex, and ulnar/radial deviation (ul/ra). Total error ($\theta_{t,tot}$) is calculated as follows:
(88)$$\theta_{t,tot}=2\cos^{-1}({\vec{q}}_{0,t,ERR})$$

For the wrist, ${\vec{q}}_{t,ERR}$ was subsequently decomposed to an Euler rotation sequence of x,y′,z″ corresponding to wrist in/ex, fl/ex, and ulnar/radial deviation (ul/ra).

RMSE for direction j is calculated as:
(89)$$\theta_{RMS,j}=\sqrt{\frac{1}{n}\sum_{t=0}^{n}\theta_{t,j}^{2}}$$

## 3. Results

Three of the six trials from one of the thirteen participants were discarded from both elbow and wrist joint angles because the forearm IMUs had shifted significantly prior to the start of the third trial. Since these two trials corresponded to a ‘medium’ and ‘fast’ transfer rate, the first trial with a ‘slow’ transfer rate was also discarded to maintain a balanced dataset across all transfer rates. Data from another participant were discarded for the wrist joint angles because the rotational alignment method failed to converge, resulting in RMS joint angles >50° for the methods that leverage the rotational constraint for the wrist.

Figure 3 shows the upper arm elevation displacements (e.g., pitch) and the corresponding elbow fl/ex for the last two trials for a participant, the rest, and the practice period between the trials. The presence of gyroscopic drift in the upper arm elevation measurements is indicated by the ‘upward’ trajectory from t = 900 s to t = 1200 s and the ‘downward’ trajectory from t = 1200 s to t = 1400, which was not present in the elbow fl/ex displacements.

RMSEs across all sensor fusion methods, motion constraints, and transfer rates are shown in Table 3 and Table 4, respectively. Total RMSE was <10.2° for the elbow across all fusion methods, motion constraints, and transfer rates. As expected, accuracy increased with the increased transfer rate. For the ‘fast’ transfer rate, the total RMS error decreased to <3.7°. Across all transfer rates and motion constraints, smoothers decreased total RMS errors to <9.2° while the map smoother decreased total RMS errors to <7.3°. The rotational motion constraint decreased total RMSE between 0.5° and 1.4° across all transfer rates for the filtering approach. It decreased total RMSE between 0.7° and 2.1° for the ‘slow’ transfer rate across the two smoothing methods. Except for the ‘slow’ transfer rate, total RMSEs were practically identical (differed by 0.1°) for all smoothers (i.e., rts-acc, map-acc, rts-dof, and map-dof) for a given transfer rate, with total errors of ~3.5° for the ‘medium’ transfer rate and ~2.0° for the ‘fast’ transfer rate. The similarities of elbow fl/ex calculated using rts-acc and rts-dof for the ‘fast’ transfer rate and the deviations during the ‘slow’ transfer rate are shown in Figure 4. Elbow fl/ex calculated using omc, mekf-acc, and map-acc for a ‘slow’ motion trial and its corresponding sample-to-sample error for one trial and one cycle are shown in Figure 5 and Figure 6, respectively.

Total RMSE was <4.4° for the wrist across all fusion methods, motion constraints, and transfer rates. As expected, accuracy increased with the transfer rate. Across all transfer rates and motion constraints, smoothers decreased total RMS errors to <2.6°. The total RMS error decreased to <2.2° for the ’fast’ transfer rate. For the wrist, the rotational motion constraint increased total RMSE between 0.4° and 1.5° for the ‘slow’ transfer rate but decreased total RMSE by 0.4° for the ‘fast’ transfer rate. Except for the ‘slow’ transfer rate, total RMSEs were practically identical (differed by 0.2°) for all smoothers (i.e., rts-acc, map-acc, rts-dof, and map-dof) for a given transfer rate, with total errors of ~1.7° for the ‘medium’ transfer rate and ~1.5° for the ‘fast’ transfer rate.

## 4. Discussion

Our results indicate that elbow and wrist kinematics can be calculated for extended durations (>30 min) using solely angular velocity and linear acceleration measurements from IMUs with RMSE <4.7° for a given motion plane for various motion speeds (i.e., dowel transfer rates). As hypothesized, for a given sensor fusion algorithm, (i) measurement accuracy improved with increased motion speeds and (ii) RMSE was smaller for the wrist than the elbow. The better performance may be attributed to greater linear accelerations at the joint center due to increased motion speeds and the longer lever arms associated with the wrist compared to the elbow.

As expected, smoothers decreased measurement error compared to the evaluated filters. The decrease in measurement error was more pronounced with the lower linear accelerations, as experienced with the elbow joint during the ‘slow’ transfer rate. During this situation, map provided an improvement over rts. Map did not produce an appreciable difference relative to rts (differed by <0.2° total RMSE) except for the elbow joint during the ’slow’ transfer rate. As hypothesized, the addition of the rotational motion constraint decreased measurement error for the elbow joint. However, in contrast to our hypothesis, adding the rotational motion constraint increased measurement errors from <2.9° to <4.4° for the wrist joint during the ‘slow’ transfer rate. This increase in error magnitudes could be attributed to the need for a more refined rotational kinematic model. It is also likely that the linear acceleration at the wrist joint provided sufficient sensor excitation for our range of testing conditions, such that the rotational constraints did not provide an added benefit. Except for the elbow joint during the ’slow’ transfer rate, a motion constraint did not produce an appreciable difference from the smoothers (differed by <0.4° total RMSE).

For the elbow joint, the relatively large variations in RMSE among the trials for the ‘slow’ transfer rate suggest that the linear acceleration at the joint center for this motion condition provided insufficient excitation for the sensor fusion algorithm for multiple participants. This limitation was mitigated through rts-acc and map-acc and the addition of a rotational motion constraint. The resulting decrease in measurement error for the ‘slow’ transfer rate is pertinent because it represents the highest error magnitude within our study and is representative of extended work durations (i.e., hours) where high motion speeds are unlikely to be sustained. While the rotational constraint did provide notable benefits for the elbow joint, the use of this constraint is (i) joint specific (i.e., a different formulation is required for the wrist) and (ii) will cause errors when the axis of constraint (e.g., the axis of ab/ad for elbow) is nearly vertical due to limited observability.

The choice of sensor fusion method is application specific. Within the context of the three different sensor fusion methods presented within this study, mekf is the only viable option if a joint angle must be computed as data are streamed. If joint angles can be calculated after collecting sensor measurements, the results indicate that rts and map can further reduce measurement error. The map sensor fusion method consistently provided the most accurate results for the elbow, especially during the ‘slow’ transfer rate. However, it did not provide any additional benefit for the wrist joint. We hypothesize that any benefits of the map sensor fusion method were not perceived for the wrist joint because our choice of material transfer rates provided sufficient sensor excitation (i.e., linear acceleration).

The goal of this study was not limited to determining the lowest error magnitudes for IMU-based motion capture but was to also understand the effects of movement speed on the error magnitude for various sensor fusion algorithms. In general, results for mekf-dof for the ‘slow’ transfer rate for the elbow were consistent with literature investigating magnetometer-free IMCs on the upper extremity. Our filter- and smoother-based results with the ‘medium’ transfer rate for the elbow were consistent with the respective filters and smoothers proposed for calculating lower extremity joint kinematics. Our results for the wrist joint and the smoothers for the ‘fast’ transfer rate provided the most accurate results among known literature.

Regarding the upper extremity, Luinge et al. [28] used a rotational constraint for the elbow joint and reported RMSE of <10° for one of the two tasks and >20° RMSE for the other. El-Gohary et al. [23] similarly used a rotational constraint for the elbow joint. They reported 6.5° and 5.5° RMSE for elbow flexion and rotation in their study involving four subjects with a recording frame of 2 min. Our study’s filter-based approach with rotation constraint (mekf-dof) offered a marginal increase in measurement accuracy for the elbow joint. RMSE <9° was observed for the elbow joint with 6.5° and 4.5° RMSE for elbow flexion and rotation. Using smoothers under increased motion speeds offered further reductions in error magnitudes. We hypothesize that our increase in measurement accuracy with mekf-dof compared to these studies may be attributed to differences in motion, the choice of kinematic model, and the model-based method for determining the rotational alignment of the IMUs. In contrast, Miezal et al. [22] reported a total RMSE of <3.5° for the shoulder, elbow, and wrist using validation from a single participant with a measurement timeframe of <30 s. The increase in accuracy in their study may be attributed to using a position-based linear motion constraint instead of the acceleration-based constraint used in this study, differences in motion, and using OMCs to derive I2S parameters.

Studies of the lower extremity may not be directly comparable due to differences in motion dynamics. Similar to the observed decrease in error magnitudes for the wrist joint compared to the elbow joint in our study, we also expect lower error magnitudes for the lower extremity than the upper extremity due to larger sensor excitations attributed to longer lever arms. Nevertheless, these studies represent the state-of-the-art magnetometer-free IMC regarding reported error magnitudes, measurement timeframe duration, and the number of participants. Weygers et al. [21] evaluated mekf-acc and map-acc for calculating knee kinematics among 11 healthy participants walking for 7 min. Mean RMSEs among the participants were <4.5° and <3.7° for each of the three motion planes for mekf-acc and map-acc, respectively. Mcgrath et al. [52] used a map smoother. They reported 3.8° RMSE for knee flexion/extension and 3.7°, 6.3°, and 4.6° for hip flexion/extension, internal/external rotation, and abduction/adduction, respectively, for 30 min of walking across 12 participants. Potter et al. [29] reported that RMSEs were generally <5° for all ankle joint angles and for flexion/extension and abduction/adduction of the hips and knee in their study of 20 participants that performed one-minute walking trials, with each trial consisting of different walking speeds as well as backward and lateral walking. Similar to our study, it was observed that RMSEs are joint and motion dependent. Left ankle dorsiflexion, for example, was observed to be 5.5° for backward walking but decreased to 4.0° for fast walking. Assuming that the motion dynamics of the lower extremity are equivalent to our elbow joint with a ‘medium’ transfer rate, our filtering approach (mekf-acc) provided improvements in measurement accuracy (<3.4° RMSE for each of the three motion planes) over these results. Teufl et al. [25] reported RMSE of <2.4° for all joints and motion planes of the lower extremity (left and right hip, knee, ankle) using a smoothing-based approach among 26 participants walking for 6 min, while our smoother approach provided equivalent error magnitudes (<2.4° RMSE for each of the three motion planes) under this same assumption.

The primary strengths of this study are (i) the number of participants and total measurement duration per participant, (ii) the use of field-capable model-based methods for determining I2S, (iii) the consideration of two smoother-based sensor fusion methods while controlling for modeling differences, and (iv) the inclusion of both elbow and wrist joint angles. The extended measurement duration is important to confirm that gyroscopic drift is eliminated. A relatively large number of participants are required to understand how variations in motion dynamics caused by individual differences will affect the performance of the sensor fusion algorithms. The use of smoothers showed the capability of this underutilized method to improve measurement accuracy. The potential benefit of the map smoother over the rts smoother was also demonstrated. Results for the elbow and wrist joints indicate that the accuracy of joint angles depends on sensor excitation, which is attributed to joint location and motion speed.

This study’s primary limitation entails using a cyclic motion pattern and lacking OMC information during rest periods. Therefore, it is unknown how well these algorithms perform for non-cyclic motion patterns and during rest periods, in which motion may be considered static or quasi-static. A secondary limitation of this study is that the IMUs were attached to the underlying segment using Velcro^®^, which could cause the sensor to shift relative to the underlying body segment over time, affecting measurement accuracy. Consistent with any sensor fusion algorithms that leverage motion constraints, soft tissue artifacts are expected to adversely affect measurement accuracy.

Future work may entail investigating the optimal conditions for conducting model-based I2S and considerations of static and quasi-static movements of the elbow and wrist, as well as validating the performance of these sensor fusion methods in static, quasi-static, and non-cyclic conditions and evaluating the computational costs of these sensor fusion algorithms.

## Figures and Tables

**Figure 1 sensors-23-07053-f001:**
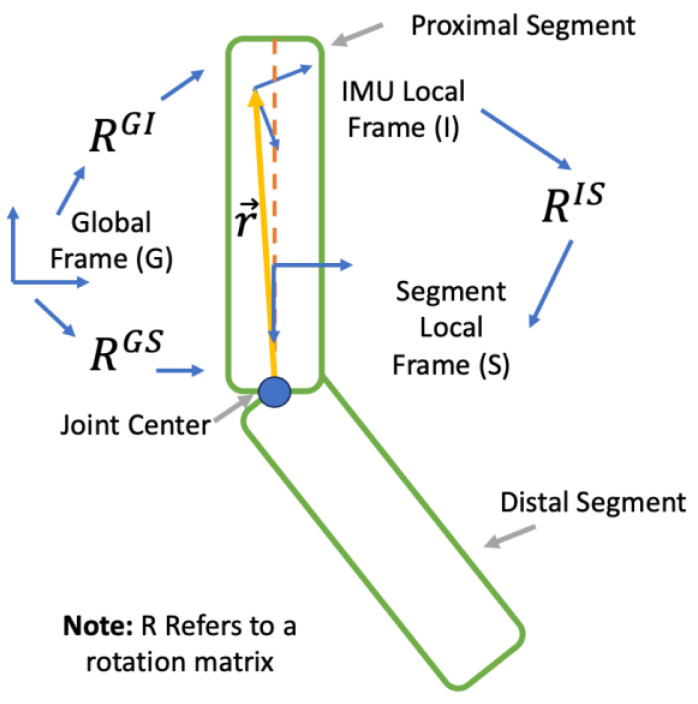
Coordinate frame definitions used within this study. *G* denotes the global frame. *I* denotes the IMU local frame. *S* denotes the segment local frame. $R^{GI}$ provides the orientation of IMU relative to Frame G, $R^{GS}$ provides the orientation of its underlying body segment relative to Frame G, and $R^{IS}$ is the offset between the IMU local frame and the local frame of its underlying body segment. $\vec{r}$ is the distance from the joint center to the position of the IMU in Frame *I*. Note that the segment coordinate frame was placed shifted in relation to the IMU local frame in the figure for legibility.

**Figure 2 sensors-23-07053-f002:**
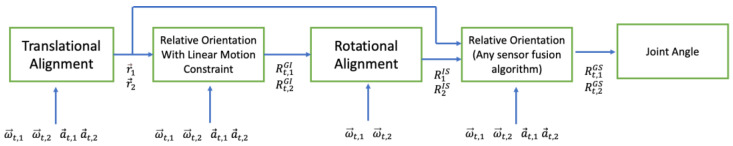
Joint angle calculation process. The translational alignment process is used to calculate the distance from the joint center to IMU attached to the proximal and distal body segment (${\vec{r}}_{i}$). This information combined with IMU linear acceleration ${\vec{a}}_{i}$ and angular velocity ${\vec{\omega }}_{i}$ measurements is used to calculate the orientation of the IMU relative to global Frame $G\left(R_{t,i}^{GI}\right)$, where *i* = 1 denotes the proximal body segment and *i* = 2 the distal body segment. The rotational offset between IMU Frame I and body segment Frame $S\left(R_{i}^{GS}\right)$ is determined from $R_{t,1}^{GI}$, $R_{t,2}^{GI}$ and ${\vec{\omega }}_{1}$, ${\vec{\omega }}_{2}$. Joint angle is subsequently calculated from ${\vec{r}}_{1}$, ${\vec{a}}_{1}$, ${\vec{\omega }}_{1}$, $R_{1}^{IS}$ and ${\vec{r}}_{2}$, ${\vec{a}}_{2}$, ${\vec{\omega }}_{2}$, $R_{2}^{IS}$.

**Figure 3 sensors-23-07053-f003:**
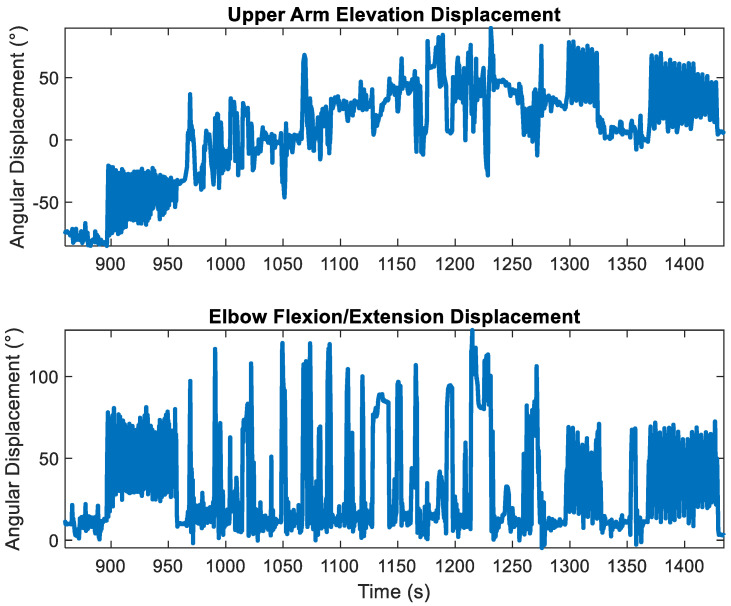
Upper arm elevation (**top**) and elbow flexion/extension (**bottom**) were calculated using the Rauch–Tung–Striebel Smoother with the rotation constraint (rts-acc) encompassing the last two trials of a participant performing the material transfer task as well as the movements during the rest period in between and the ‘practice’ period. Time is in seconds elapsed since the beginning of the first trial. Note the presence of gyroscopic drift in the upper arm elevation measurements, as indicated by the ‘upward’ trajectory from t = 900 s to t = 1200 s and the ‘downward’ trajectory from t = 1200 s to t = 1400, that was not present in the elbow flexion/extension displacements.

**Figure 4 sensors-23-07053-f004:**
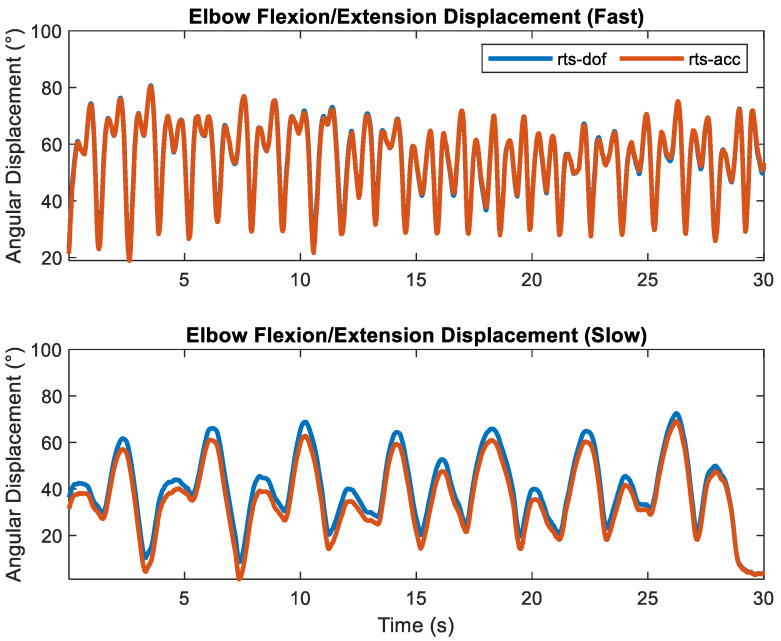
Elbow flexion/extension for the ‘fast’ transfer rate (**top**) and ‘slow’ transfer rate (**bottom**) calculated using the Rauch–Tung–Striebel Smoother with the rotation constraint (rts-dof) and without (rts-dof).

**Figure 5 sensors-23-07053-f005:**
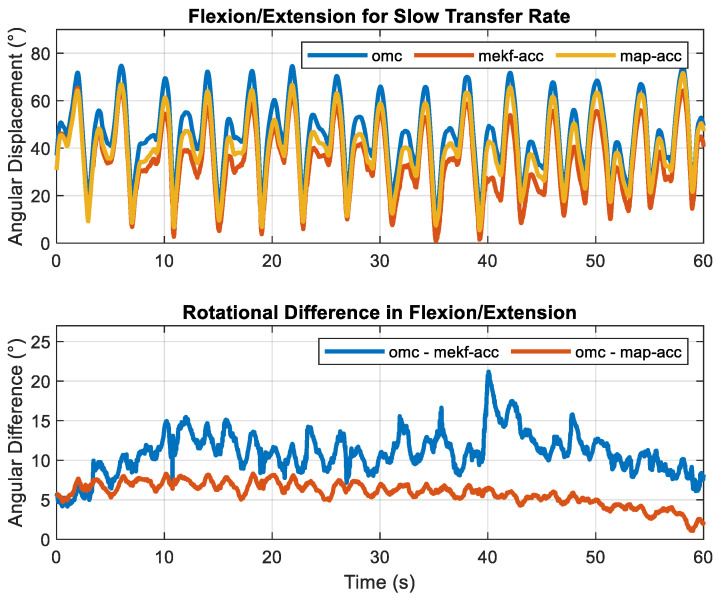
Elbow Flexion/Extension for a single one-minute trial with a ‘slow’ transfer rate calculated using an optical motion capture system (OMC), linear acceleration-based kinematic constraint, and its sample-to-sample difference calculated using two different sensor fusion algorithms: a Multiplicative Extended Kalman Filter (mekf-acc) and Maximum A Posteriori Smoother (map-acc).

**Figure 6 sensors-23-07053-f006:**
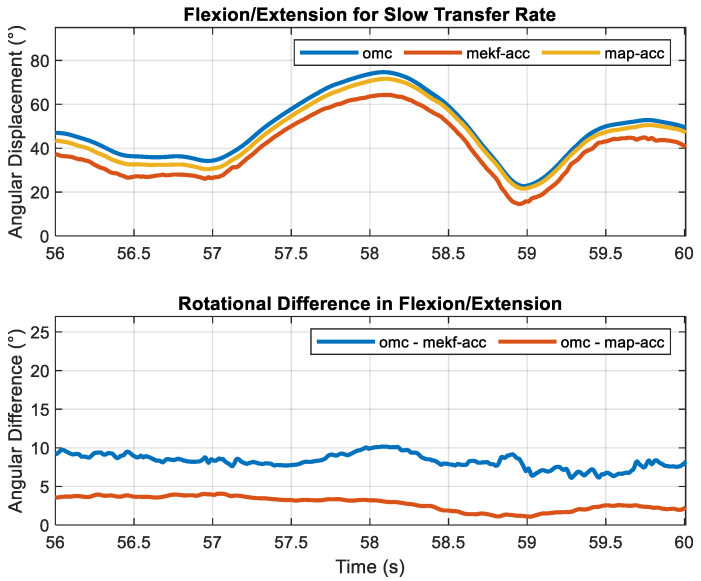
One cycle of elbow Flexion/Extension for a single trial one minute in length with a ‘slow’ transfer rate calculated using an optical motion capture system (OMC), linear acceleration-based kinematic constraint, and its sample-to-sample difference calculated using two different sensor fusion algorithms: a Multiplicative Extended Kalman Filter (mekf-acc) and Maximum A Posteriori Smoother (map-acc).

**Table 1 sensors-23-07053-t001:** Filter parameters for the elbow joint consisting of Initial uncertainty P_0_ and standard deviation of the gyroscope noise $\sigma_{\omega }$, linear acceleration constraint $\sigma_{acc}$, and rotational motion constraint $\sigma_{dof}$ for the multiplicative Extended Kalman Filter (mekf), Rauch–Tung–Striebel Smoother (rts), and Maximum A Posteriori Smoother (map) with linear acceleration kinematic constraint (acc) and with the degree of freedom constraint in addition to linear acceleration constraint (acc-dof). Note $\sigma_{dof}$ is unitless.

	mekf-acc	rts-acc	map-acc	mekf-dof	rts-dof	map-dof
**P_0_**	$WQW^{T}$	$WQW^{T}$	diag([1, 1, 1])	$WQW^{T}$	$WQW^{T}$	diag([1, 1, 1])
$\sigma_{\omega }\;$ **(rad/s)**	0.005	0.005	0.005	0.005	0.005	0.005
$\sigma_{acc}\;(m/s^{2})$	0.01	0.02	0.04	0.05	0.05	0.05
$\sigma_{dof}$	-	-	-	0.01	0.02	0.04

**Table 2 sensors-23-07053-t002:** Filter parameters for the wrist joint consisting of initial uncertainty P_0_ and standard deviation of the gyroscope noise $\sigma_{\omega }$, linear acceleration constraint $\sigma_{acc}$, and rotational motion constraint $\sigma_{dof}$ for the multiplicative Extended Kalman Filter (mekf), Rauch–Tung–Striebel Smoother (rts), and Maximum A Posteriori Smoother (map) with linear acceleration kinematic constraint (acc) and with the degree of freedom constraint in addition to linear acceleration constraint (acc-dof). Note $\sigma_{dof}$ is unitless.

	mekf-acc	rts-acc	map-acc	mekf-dof	rts-dof	map-dof
**P_0_**	$WQW^{T}$	$WQW^{T}$	diag([1, 1, 1])	$WQW^{T}$	$WQW^{T}$	diag([1, 1, 1])
$\sigma_{\omega }\;$ **(rad/s)**	0.005	0.005	0.005	0.005	0.005	0.005
$\sigma_{acc}\;(m/s^{2})$	0.01	0.02	0.04	0.05	0.05	0.05
$\sigma_{dof}$	-	-	-	0.04	0.08	0.08

**Table 3 sensors-23-07053-t003:** Mean (SD) flexion/extension, abduction/adduction, pronation/supination, and total root mean square error (RMSE) for the elbow joint calculated using Multiplicative Extended Kalman Filter (mekf), Rauch–Tung–Striebel Smoother (rts), and Maximum A Posteriori Smoother (map) with linear acceleration kinematic constraint (acc) and with the degree of freedom constraint in addition to linear acceleration constraint (acc-dof).

	mekf-acc	rts-acc	map-acc	mekf-dof	rts-dof	map-dof
**Slow**						
Flexion	7.3(4.4)	7.0(7.4)	5.2(3.2)	6.5(4.4)	5.2(3.2)	4.7(2.7)
Adduction	4.5(3.2)	3.3(2.5)	3.0(1.8)	3.8(2.1)	3.0(2.0)	2.8(2.0)
Rotation	5.2(3.3)	4.6(4.1)	3.8(2.2)	4.2(2.4)	3.4(1.8)	3.2(1.5)
Total	10.2(6.1)	9.2(8.3)	7.3(3.9)	8.8(5.0)	7.1(3.7)	6.6(3.4)
**Medium**						
Flexion	3.4(1.6)	2.3(1.1)	2.4(1.2)	3.2(1.5)	2.4(1.1)	2.4(1.1)
Adduction	2.5(1.4)	1.6(0.8)	1.5(0.7)	2.0(1.0)	1.6(0.7)	1.5(0.6)
Rotation	2.9(1.3)	1.9(0.7)	2.0(0.9)	2.7(1.6)	2.0(1.0)	2.0(0.8)
Total	5.3(2.2)	3.5(1.4)	3.5(1.4)	4.8(2.1)	3.6(1.4)	3.6(1.3)
**Fast**						
Flexion	2.3(0.5)	1.2(0.3)	1.2(0.4)	1.9(0.6)	1.2(0.4)	1.2(0.4)
Adduction	1.9(0.7)	0.9(0.3)	1.0(0.3)	1.4(0.4)	0.9(0.3)	1.0(0.3)
Rotation	2.0(0.7)	1.1(0.3)	1.2(0.4)	1.7(0.7)	1.1(0.3)	1.2(0.4)
Total	3.7(1.0)	1.9(0.4)	2.0(0.5)	3.0(0.8)	1.9(0.5)	2.0(0.5)

**Table 4 sensors-23-07053-t004:** Mean (SD) Internal/external, flexion/extension, ulnar/radial deviation, and total root mean square error (RMSE) for the wrist joint calculated using Multiplicative Extended Kalman Filter (mekf), Rauch–Tung–Striebel Smoother (rts), and Maximum A Posteriori Smoother (map) with linear acceleration kinematic constraint (acc) and with the degree of freedom constraint in addition to linear acceleration constraint (acc-dof).

	mekf-acc	rts-acc	map-acc	mekf-dof	rts-dof	map-dof
**Slow**						
Rotation	1.8(0.6)	1.5(0.5)	1.6(0.5)	2.5(0.9)	1.7(0.5)	1.6(0.5)
Flexion	0.8(0.3)	0.6(0.2)	1.0(0.4)	1.3(0.9)	0.7(0.3)	1.0(0.4)
Deviation	2.0(1.3)	1.4(0.9)	1.5(0.6)	3.4(1.6)	1.7(0.7)	1.5(0.6)
Total	2.9(1.3)	2.2(0.9)	2.4(0.6)	4.4(1.9)	2.6(0.8)	2.5(0.7)
**Medium**						
Rotation	1.5(0.6)	1.2(0.5)	1.2(0.5)	1.5(0.5)	1.2(0.5)	1.2(0.5)
Flexion	0.8(0.3)	0.6(0.2)	0.8(0.2)	0.8(0.4)	0.6(0.2)	0.8(0.2)
Deviation	1.5(1.0)	0.8(0.6)	1.1(0.5)	1.4(0.7)	0.8(0.5)	1.0(0.5)
Total	2.3(1.1)	1.7(0.7)	1.8(0.6)	2.3(0.8)	1.6(0.5)	1.8(0.6)
**Fast**						
Rotation	1.4(0.5)	1.1(0.3)	1.1(0.4)	1.2(0.3)	1.0(0.3)	1.0(0.4)
Flexion	1.0(0.5)	0.8(0.2)	0.8(0.3)	0.8(0.2)	0.7(0.2)	0.8(0.3)
Deviation	1.3(0.6)	0.7(0.3)	0.9(0.3)	1.0(0.4)	0.6(0.2)	0.8(0.2)
Total	2.2(0.9)	1.5(0.4)	1.6(0.4)	1.8(0.4)	1.4(0.3)	1.6(0.4)

## Data Availability

The data are not publicly available due to the human subject-derived nature of the dataset.

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
