# Peer review of "Drift-Free Joint Angle Calculation Using Inertial Measurement Units without Magnetometers: An Exploration of Sensor Fusion Methods for the Elbow and Wrist"

_sensors, 2023, doi:10.3390/s23167053_

Round 1

Reviewer 1 Report

The paper presents a study of human motion analysis with an IMU, and some inertial sensor data processing methods are proposed. Some experimental results are given to support the advantages or feasibility of the proposed sensing system. However, this kind of motion captures are commonly used in the human analysis in the recent human motion analysis, so the important innovation points should be clarified.

The main innovation idea for the paper should be clarified. In the method section, some algorithms explanation figures should be provided, too many equations are difficult to explain that the acceleration errors or noise are removed to calculate joint angles of upper limbs.

The experimental method should be explained with more detail information, and how to validate the effects of the sensors miss alignment to the human movements measurements?

The discussion section should be improved with more publication references. Some related papers recently published should be referred. The obtained results should be compared in a list form related the other research.

The English is pretty good.

Reviewer 2 Report

This work claims three theoretical contributions in the application of wearable sensors to estimate the upper limb (elbow and wrist) joint angles, and then presents a comparison of a number of sensor fusion techniques for IMUs without magnetometer data. In total 6 methods in total were compared, using combinations of Multiplicative Extended Kalman Filter, a Rauch-Tung-Striebel Smoother, and a Maximum A Posteriori sensor fusion algorithms with and without the use of rotational constraints related to the modelled degrees of freedom.

The work presents the mathematical details of each of these approaches, and compares the performance against an optical motion capture system with 13 participants in a material transfer task carried out at three different speeds. 

Overall the work seems like a reasonable comparison between algorithms using a given dataset in a useful application. However, the presentation of the work makes it difficult to understand and evaluate the contributions claimed. The following comments detail areas in which the work could be improved. 

1. The presentation of the algorithms is at times difficult to follow. Details examples are given at the end of the review (see "Notes on Mathematical Presentation"), but I suggest that the presentation may be improved in one of the following ways:

(1) If possible, providing an overview of the algorithms in graphical (possibly flowchart) or pseudocode form, before the equations are presented, such that it is easier to follow

(2) If there is a significant overlap with existing works, only describing how the proposed method differs from the state of the art and instead providing a description of the existing algorithms only. 

2. It would be easier to understand with a diagram of the limb segments and sensors being used, along with the frames defined. Particularly, this should include an indication of which limb segments are being measured, and which are the degrees of freedom allowed in the rotational constraint. This would greatly improve the readability of the work. 

3. It is not clear why the magnetometer information is not used - or why it cannot be used in these methods. The authors refer to one of their previous works to justify the large errors ([4]), however, that work describes a drift with respect to a filter being applied to a single sensor. It does not (to this reviewer) present a convincing argument to completely ignore the information provided by the magnetometer, and it would be interesting to discuss how this would affect the the kinematic estimation - especially given the enforced proximity of the sensors on the arm. 

4. The contribution and presentation of the IMU-to-segment alignment is confusing. It is presented as a distinct contribution (and appears to be a modification of an existing technique which does include 'disturbed' magnetometer readings), but is presented after the algorithms in the methods. However, it would appear that this alignment should occur before the application of the algorithms, and only once (and commonly) for all algorithms. This, however, could be clearer in the text. Additionally, the results do not report the accuracy of this alignment method, nor compare it to the existing state of the art. It would be useful if a more complete presentation of this contribution is included in the work, or, if it is less emphasized in the introduction and methods. 

5. Similarly, the authors claim a description of a 'computationally efficient' RTS smoother formulation in the introduction, however, it is not clear how the presentation here is different to the state of the art, nor why it is computationally efficient. This again should be clarified.

6. With respect to the three points above, due to these complexities and number of contributions claimed, it would be useful if the results and discussions were clearly identified against these contributions. 

7. It is not clear why the \sigma listed in Tables 1 and 2 would change for the different algorithms. Typically this parameter is a covariance which is dependent on the noise properties of the sensor, and thus should be consistent across approaches.

8. The discussion states "The goal of this study was not to provide the lowest error magnitudes for IMU-based motion capture but to understand the effects of movement speed on the error magnitude for various sensor fusion algorithms." However, this is not at all clear from the introduction, with the exception of a brief mention at the end. If this is indeed the goal of the work, some discussion should be included in the introduction as to why one would expect different results with the different algorithms. 

9. A large paragraph in the discussion is devoted to reporting existing results. However, it is not clear how they relate to the present work, particularly the ones associated with lower limb measurements, for which no attempt to provide equivalence to the movements in this study is provided. 

10. Given the repetitive nature of the task, a discussion on how the results would relate to more generic situations, or other movements, is required to provide interest to a greater audience. 

Notes on Mathematical Presentation:

The presentation of the mathematics is at times difficult to follow - occasional terms are not well defined or signposted, and the notation is not consistent.  For example:

- Equations for the covariance P_{t|t-1} are defined in (11), but the state to which the covariance is related to is not defined, nor the dimension

- The notation of K_t and K_{t,i} is confusing, given that they do not appear to share a common meaning

- Describing the mathematics "without the rotational constraint" in (19) to (23) is confusing given the introduction of the rotational constraint much later in the manuscript.

Additional minor notes:

- It's not clear what \phi_1, \phi_2 refer to in (71). Perhaps they should be \psi?

- It's not clear what reference orientation \alpha_r is referring to in (79)

Reviewer 3 Report

The approach is interesting and the presentation of the methodology is good. But the interest of the paper is in the results and the conclusions derived from them, as proposing novel methodologies is not the aim of the paper. Therefore, my main suggestions refer to the need of improving the Results and Discussion sections.

SPECIFIC COMMENTS

·         In the Introduction section, the authors should limit the Introduction section to clearly and concisely justifying the motivation for their work, based on existing knowledge, and to defining the objectives and starting hypotheses. The review of the state of the art should be moved to the discussion section. In this review, the previous published results should be compared with those obtained in this work, and the advances achieved by the authors should be highlighted.

·         Some acronyms appear that have not been previously defined (acc) and, on the contrary, some previously defined acronyms are redefined (e.g. in the titles of some tables). Furthermore, these acronyms are sometimes in upper case (MEKF) and sometimes in lower case (mekf). Please correct

·         Line 226, editing error

·         The movement of the IMU with respect to the soft tissue and the deformation of the soft tissue itself affects the detection of the movement of the bony elements of the joints.  Please discuss on this source of error. Aditionally, was there any procedure to check that the IMUs, fixed with Velcro straps, had not moved during the tests?

·         I have several concerns about the use of Optitrack. Was it used as the control measurement system? I.e. with respect to which the errors of the IMUs are compared? If so, the accuracy of the angular measurement after calibration of the system must be provided (camera resolution is not enough). If this was not the purpose , the aim of using  the OMC data in this work should be indicated.

·         Please explain how the IMU position is calculated when using the OMC system. It is said that 4 markers are placed on its surface, but this information is redundant since the position of a rigid solid is defined by 3 points (if they are not aligned).

·         Table 3 and 4, show the mean angular displacement over each trial? If so, what is the usefulness of these mean values? I find the range of variation of each variable more meaningful in order to assess the impact of the error on the measurements. Furthermore, you provide the mean values of the angular displacements, but when the mean RMS is greater than (mean±SD) of the measured variables (in tables 3 and 4), would it imply that such measurement system is not sufficiently accurate and, therefore, the results lack significance? Please discuss this issue

·         Furthermore, you provide the mean values of the angular displacements, but when the mean RMS is greater than [mean+-SD] of the measured variables (tables 3 and 4), would it imply that such a measurement system is not sufficiently accurate and therefore the results lack significance? Please discuss this issue

Round 2

Reviewer 2 Report

Thank you for your revision of the paper. On the whole it is greatly improved. However, there are still some issues to be addressed.

1. In making the revisions, the paper (and the response letter) has not been appropriately proofread. There are a number grammatical issues; very long flow-on sentences which are hard to follow; missing references; and a comment from the previous review on line 143, which appears to have been copied into the text and forgotten about. The authors should ensure that they are appropriately vigorous in their submission to demonstrate care and respect to the editorial process. 

2. The paper would be greatly improved by a more hypothesis-driven presentation of the algorithms and the results. For example, two "smoothing" algorithms are represented in the work (RTS, MAP), with the MAP presented as having "more flexibility and accuracy... at the expense of computational costs". As such, this trade off should be explored in the results and the discussion (noting that no results are presented on the computational cost) - thus it is hard for the reader to be convinced or informed of the actual trade-off being made. 

3. Similarly, the discussion notes the difference in performance due to fast/slow transfer rates, with different algorithms performing differently at different transfer rates. However, it is not clear in the prior presentation that this is something which may potentially be a differentiating factor in the relative performance of the algorithms, nor are any hypotheses formally offered with respect to this. Again, the contribution and significance of the work could be greatly improved if the work was rephrased around these measurements. 

4. Comment 9 from the previous study has not been sufficiently addressed. The paragraph in the discussion starting on Line 622 (regarding the lower limb results) still simply lists results. Whilst there is likely some relevance, this discussion should guide the reader through what this means in the context of the results of THIS paper. Are the authors arguing that the results are comparable? Do measurements of the upper limb have higher/lower acceptable levels of error? Providing a more clear discussion would greatly aid in improving the significance of this work. 

5. Have the authors considered presenting the results (particularly Tables 3 and 4) as a 2-dimensional matrix (perhaps even as a 3-D plot of some representation)? It appears that algorithms under test are a combination of two variables (filtering algorithm and acc/dof), and thus presentation in this way may make it easier to see the split the effects of the algorithm and the dof-constraint).

Please proofread more carefully before submission. 
